palaeontology/evolution

Squamata, ecomorphology, macroevolution, Cretaceous Terrestrial Revolution, palaeontology

**Author for correspondence:**
Michael J. Benton
e-mail: mike.benton@bristol.ac.uk

# Ecomorphological diversification of squamates in the Cretaceous

Jorge A. Herrera-Flores, Thomas L. Stubbs and Michael J. Benton

School of Earth Sciences, University of Bristol, Life Sciences Building, Tyndall Avenue, Bristol BS8 1TQ, UK

 JAH-F, 0000-0002-9660-4161; TLS, 0000-0001-7358-1051; MJB, 0000-0002-4323-1824

Squamates (lizards and snakes) are highly successful modern vertebrates, with over 10 000 species. Squamates have a long history, dating back to at least 240 million years ago (Ma), and showing increasing species richness in the Late Cretaceous (84 Ma) and Early Palaeogene (66–55 Ma). We confirm that the major expansion of dietary functional morphology happened before these diversifications, in the mid-Cretaceous, 110–90 Ma. Until that time, squamates had relatively uniform tooth types, which then diversified substantially and ecomorphospace expanded to modern levels. This coincides with the Cretaceous Terrestrial Revolution, when angiosperms began to take over terrestrial ecosystems, providing new roles for plant-eating and pollinating insects, which were, in turn, new sources of food for herbivorous and insectivorous squamates. There was also an early Late Cretaceous (95–90 Ma) rise in jaw size disparity, driven by the diversification of marine squamates, particularly early mosasaurs. These events established modern levels of squamate feeding ecomorphology before the major steps in species diversification, confirming decoupling of diversity and disparity. In fact, squamate feeding ecomorphospace had been partially explored in the Late Jurassic and Early Cretaceous, and jaw innovation in Late Cretaceous squamates involved expansions at the extremes of morphospace.

## 1. Introduction

Extant squamates, represented by lizards, snakes and amphisbaenians, are one of the most successful groups of living vertebrates with a diversity of over 10 200 living species [1]. This diversity is matched by a great range of dietary modes, including herbivory, insectivory and carnivory, sometimes associated with the use of venom, as well as more specialized diets such as seagrass-eating and mollusc-eating [1].

Squamates have existed on Earth for 240 million years (Myr) or more. Their fossil record is relatively sparse for the first half of their history [2–7]. The oldest 'squamate' is *Megachirella wachtleri* from the Middle Triassic of Italy [7], a member of the stem group and outside all living squamate clades. The oldest known true squamates are fragmentary remains from the Middle Jurassic of England, including representatives of most key modern orders such as gekkotans, scincomorphs, anguimorphs, iguanians and snakes [2,8,9]. The fossil record [2,3,7–9] and phylogenomic studies [4–7] point to an increase in diversity of squamates in the Late Cretaceous, some 84 Ma. This diversification also included a major marine group, the mosasauroids, which became ecologically abundant predators in shallow seas around the world, and some reached huge sizes, before their extinction at the end of the Cretaceous [10]. Modern squamate clades continued to diversify through the Late Cretaceous and in the Palaeogene, after the end-Cretaceous mass extinction 66 Ma. Phylogenomic analyses and lineages-through-time plots [4–7] show continuing diversity rise from about 84 Ma and again in the Palaeogene; however, the fossil record of Palaeogene squamates is sparse, probably under-sampling this biodiversity [3]. Our question is whether the fossil data document a parallel rise in dietary ecomorphological disparity through the Cretaceous, or whether diversity and disparity are decoupled, as has commonly been observed in fossil examples, when disparity commonly increases before diversity [11].

In order to explore this question, we focus on the Mesozoic fossil record of squamates, the first two-thirds of their history. The diversity history of squamates has been explored before, based both on the fossil record [2,3] and on phylogenomic analyses [4–7], which show very low diversity from the Triassic to Mid-Cretaceous, and then bursts of diversity in the Late Cretaceous and Palaeogene. Our approach is to focus instead on morphological disparity, the range of anatomical form. Morphological disparity can be as a proxy for ecological variety, either by measuring aspects of the skeleton for which feeding or other functions can be assigned or by linking morphologies with modern taxa whose habits are known [8,12,13]. Here, we explore dietary ecomorphology to understand the changing diversity and disparity of squamates through the first three-quarters of their history. This enables us to use the richest available data from the fossil record of squamates; for many taxa, tooth-bearing mandibles, premaxillae and maxillae are the only, or best-preserved, elements. Further, the teeth and jaws of lizards can represent their diets, discriminating herbivores from carnivores, but also more specialized feeding modes [2]. We follow earlier work [2,8,13,14] that established how tooth shape in modern squamates is linked to diet, and that these observations can be extended back to assign likely diets to Mesozoic squamates. We enrich these data with an exploration of jaw size and shape, which are also important ecomorphological proxies often linked to dietary guilds.

It is not at all clear whether diversity and disparity should evolve in concert, perhaps a null expectation [11], or whether they are decoupled. Here, we examine three key data resources: dental disparity, jaw sizes and lower jaw shape disparity. We find that all metrics agree that a substantial expansion of squamate ecomorphological disparity occurred 15–20 Myr before the first rise in taxonomic diversity of the clade in the Campanian. The three indices provide different insights into the drivers, timings and magnitudes of ecomorphological innovation. It seems that ecomorphological disparity was decoupled from species richness, and squamate ecological disparity expanded at least 25 Myr before diversity.

## 2. Methods

### 2.1. Dental disparity

We compiled a database of dental morphotypes for 220 Mesozoic squamate genera. Generic occurrence records for all squamates ranging from the Triassic to end-Cretaceous were downloaded from the Palaeobiology Database (PBDB; www.paleobiodb.org), accessed via Fossilworks (www.fossilworks.org) in October 2020. We could not include squamates before the Late Jurassic as occurrences are scarce and sporadic, and the fossils are too incomplete to show jaw and dental characters sufficiently. We stopped sampling at the end of the Cretaceous, as we wanted to keep the focus simply on the pre-mass extinction interval. We chose to work at the generic level because there is little intrageneric variation in the ecomorphological traits we consider, particularly as most Mesozoic squamate genera are monospecific, so our analysis is effectively at the species level. Full details are given in the electronic supplementary material.

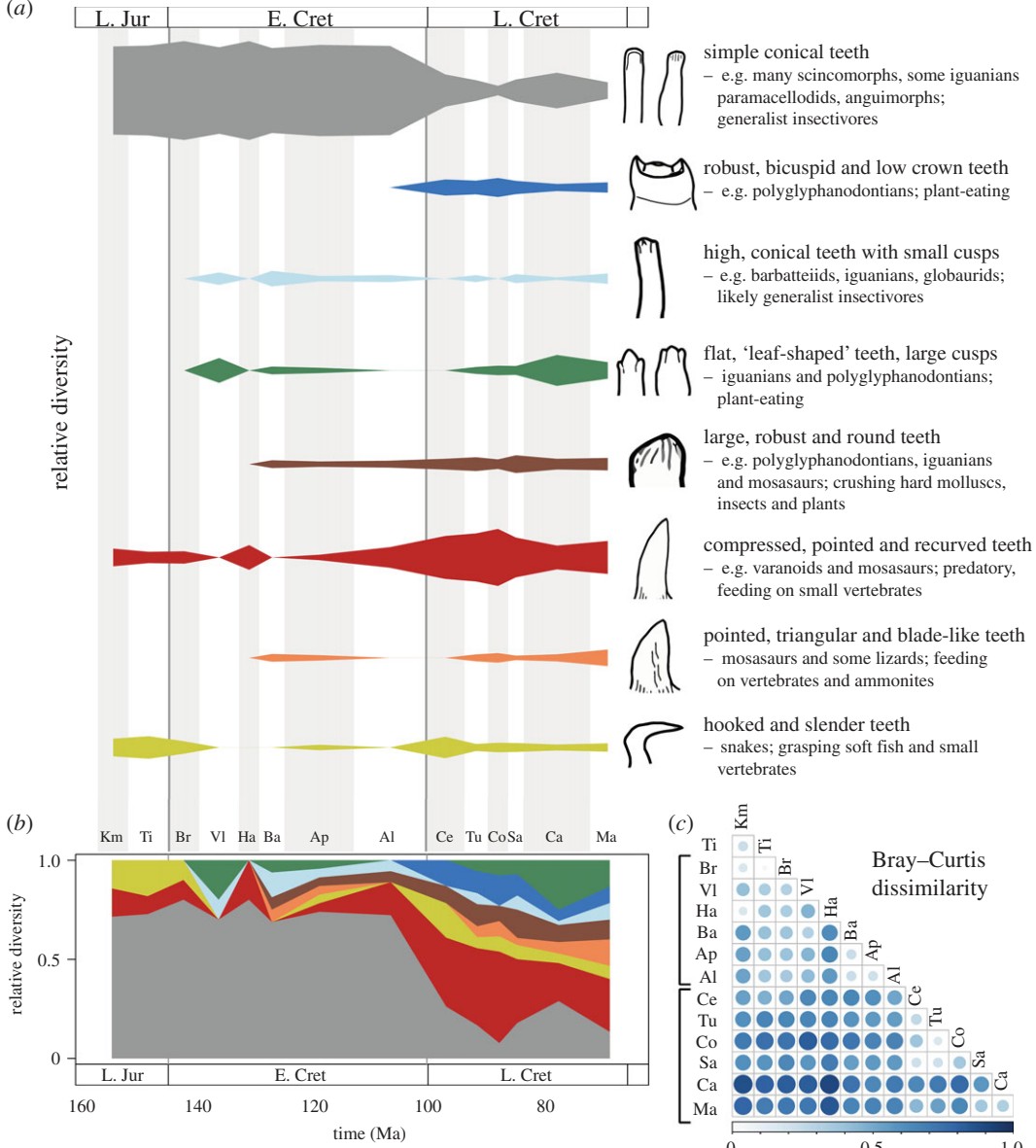

**Figure 1.** Dental disparity in Mesozoic squamate genera ($n = 220$). (*a*) The relative (proportional) diversity of eight tooth morphotypes identified among modern and Mesozoic squamates, in 14 stage-level bins from the Late Jurassic to end-Cretaceous. For each tooth morphotype, information is provided on their general phylogenetic occurrences and broad diets as observed among living and fossil forms. (*b*) Stacked relative (proportional) diversity of the eight tooth morphotypes through time. (*c*) Pairwise comparisons of Bray–Curtis dissimilarity for dental occurrences in the 14 geological stages. Large circles and darker blue shading indicate greater dissimilarity in the dental occurrences between bins. Results excluding marine taxa and based on the absolute numbers of occurrences are presented in electronic supplementary material, figures S1 and S2.

Taxa were assigned to dental morphotypes in eight general categories (figure 1*a*). These were determined from the literature [13,14], focusing on the original descriptions of taxa, and using photographs, written descriptions and drawings of individual specimens. The dental categories were defined to encapsulate the full diversity of dental morphologies present in the squamate fossil record, and they can all be linked with modern squamates whose diets are known [13,14]. Individual tooth shapes do not always indicate a precise diet, and the best we can do is to compare with modern taxa with identical morphotypes and make a broad inference about diet. Taxa showing heterodonty were assigned to more than one dental morphotype, as appropriate based on their varied tooth shapes. Temporal trends in the proportions of dental morphotypes were examined by calculating the relative diversity of each morphotype in 14 geological stages (Kimmeridgian to Maastrichtian) with ranged-through bin sampling. We calculated pairwise Bray–Curtis dissimilarity between the stage level bins

to explore changing dental disparity through time using the R package ecodist [15] and plotted results using corrplot [16].

## 2.2. Size evolution

Size disparity was assessed by using lower jaw length as a proxy. In summary, we find that mandible length is a good proxy for skull size, but not for overall size. In the face of a fragmentary fossil record, Mesozoic squamate lower jaws are the most commonly preserved complete elements. Further, mandibles are usually diagnostic of genera, so we could focus on named taxa only, not taxonomically unassigned specimens. Mandible length has been used previously as a proxy for overall body size [2], and its use allows us to maximize the size of the dataset. It is important to consider that squamates do show differences in jaw size relative to total body length (and likely body mass). For example, snakes and amphisbaenians typically have highly elongate bodies compared with their jaw length, while some lizards (e.g. skinks and geckos) have relatively longer jaws compared with total body length [1]. Nevertheless, we consider jaw size an important component of ecomorphological variation and we are interested in large-scale trends incorporating great magnitudes of size disparity, ranging from tiny lizards with jaws less than 10 mm, moderately large varanoids and polyglyphanodontians with jaws around 10 cm, to giant mosasauroids with jaws over 1.5 m.

A database of lower jaw lengths for 116 genera was compiled, representing all complete specimens available, and we used the maximum jaw length of the largest known specimen confidently referable to each taxon (see electronic supplementary material). Lower jaw lengths were taken directly from specimens, the literature or measured from pictures using ImageJ [17]. We explored temporal trends of size evolution by plotting $\log_{10}$-transformed lower jaw length against geological time based on the stratigraphic ranges of all taxa. We used permutation tests to identify significant shifts in within-bin size disparity based on univariate range and standard deviation. We used multi-stage and epoch level bins for comparisons because of small sample sizes. The permutation tests compare the observed difference in size disparity between two bins to that expected if group memberships in those bins were randomized, based on 1000 iterations.

## 2.3. Lower jaw disparity

We studied changes in squamate morphospace occupation through the Mesozoic based on variation in lower jaw shape. Lower jaw shape is a commonly used ecomorphological proxy, because shape innovations are linked to dietary specializations in tetrapods [12,18–20]. We compiled a database of two-dimensional images of lower jaws, oriented all images to the same side (right), and set seven fixed landmarks and 26 semi-landmarks on the lower jaw images (figure 3b), using tpsDig [21]. Before performing principal components analysis (PCA), we carried out a generalized Procrustes analysis to correct for variable size, positioning and orientation of the specimens. All corrected coordinates were then subjected to PCA in R [22], using the package geomorph [23]. Two analyses were performed, one incorporating all samples (n = 89) to explore temporal trends in the Late Jurassic (n = 5), Early Cretaceous (n = 13) and the Late Cretaceous (n = 74), and a separate analysis for only the well-sampled Late Cretaceous taxa to explore the distribution of higher groups (e.g. lizards, snakes, mosasaurs, etc.) and dietary groups. Dietary groups for Late Cretaceous squamates were inferred by tooth morphology or by suggested diets provided in the literature (see electronic supplementary material).

Within-bin disparity was calculated for epoch temporal bins based on the sum of variances metric using all morphospace axes, and the convex hull volume metric based on PC1–PC3. For the sum of variances metric, bootstrapping with 500 iterations was used to generate 95% confidence intervals. As the convex hull volume metric is susceptible to sample size differences, we applied rarefaction to this metric for the Late Cretaceous bin, making sample size match the preceding Early Cretaceous bin (n = 13) and incorporating bootstrapping with 500 iterations to generate 95% confidence intervals. Permutation tests were again carried out to identify significant differences in within-bin epoch disparity based on the sum of variances and convex hull volume metrics. Non-parametric multivariate analyses of variance (NPMANOVA) were applied to all PC axes to test for significant differences in morphospace positioning in epoch bins. Disparity calculations were performed using the R package dispRity [24] and custom code (see electronic supplementary material).

Our jaw shape sampling incorporates all major taxonomic groups in the bins they are known, with one notable omission; snakes from the Late Jurassic and Early Cretaceous. Snake origins have been traced back tentatively to the Middle Jurassic [9], but no suitable material for landmarking is known

until the Late Cretaceous. Omitting snakes from the Late Jurassic and Early Cretaceous bins may underestimate squamate disparity during these times. To circumvent this, we incorporated a snake 'morphotype' within those bins based on the average shape of the Late Cretaceous snakes we sampled. This 'average' snake morphology was then considered when exploring morphospace occupation and calculating disparity in the Late Jurassic and Early Cretaceous in a second series of analyses.

# 3. Results

## 3.1. Dental disparity

Trends of dental disparity show a marked shift from a homogeneous assemblage dominated by plesiomorphic conical tooth forms in the Late Jurassic and Early Cretaceous, to a more heterogeneous assemblage including more complex teeth in the Late Cretaceous. Early squamates, specifically Late Jurassic (Kimmeridgian–Tithonian) taxa, had low dental disparity, essentially comprising three morphotypes (simple conical; compressed, pointed and recurved; hooked and slender), and highly dominated by taxa with simple conical teeth lacking notable specializations. These simple forms are often described as 'chisel-like' and 'peg-like', with minor apical faceting or striations (figure 1). In the Early Cretaceous, new dental morphotypes appeared, including those with increasing cuspidy and crushing adaptations, but taxa with simple conical teeth were still most abundant, comprising 68–80% of occurrences (figure 1). In the Cenomanian, there was a clear turnover in the dental disparity of squamates, with much of the increased disparity of other tooth types initially occurring within a fixed, overall low diversity of species, before the massive sampling/diversity increase in the Campanian. The relative proportion of taxa with conical teeth declined substantially, while predatory morphotypes with pointed and recurved teeth showed a large increase in relative diversity (figure 1). During the Late Cretaceous, there was also increased relative diversity of other rarer and more complex dental morphotypes, including distinct, transversally bicuspid, forms, taxa with labiolingually compressed 'leaf-shaped' teeth with notably increased cuspidy (tricuspid and polycuspid), and taxa with robust crushing dentitions (figure 1). Pairwise comparisons based on Bray–Curtis dissimilarity highlight the stepwise changes in dental disparity, with high dissimilarity scores for all Late Cretaceous stages when compared with the Early Cretaceous, and high dissimilarity between the Campanian and Maastrichtian bins compared with the Cenomanian–Santonian interval (figure 1c). If marine squamates are excluded, the same general patterns are recovered. The relative diversity of pointed and recurved teeth in the Late Cretaceous is greatly reduced, but there is still a shift in the proportions of simple conical teeth compared with more complex and specialized forms in the Late Cretaceous, beginning in the Cenomanian (see electronic supplementary material, figure S1).

## 3.2. Size evolution

During most of early squamate evolution, the group was characterized by small size (figure 2). From the Oxfordian to the Albian, squamates included taxa of small to moderate size, with lower jaw lengths less than 100 mm. There is a significant increase in the range and spread of jaw sizes into the Cenomanian and Turonian (Cenomanian–Turonian bin versus Aptian–Albian bin: observed versus expected range $p = 0.038$, observed versus expected standard deviation $p = 0.016$; electronic supplementary material, figure S3). This interval was associated with a significant increase in mean jaw size (Cenomanian–Turonian bin versus Aptian–Albian bin: Welch two sample $t$-test, $t = -4.2484$, d.f. = 25.144, $p = 0.0003$). Greatest disparity in lower jaw sizes is seen in the Campanian, when squamate jaws ranged from approximately 10 mm to approximately 1700 mm long. Maastrichtian taxa showed a very similar range of sizes to the Campanian.

The Middle Cretaceous rise in jaw size, and the associated increase in the range and spread of sizes, is driven by the diversification of marine squamates, including dolichosaurs, snakes and early mosasauroids (figure 2a). Mosasauroids dominate the large body sizes, with jaws from 200 mm to over 1 m long (figure 2a). It is difficult to compare terrestrial squamates between the Aptian and Albian on the one hand with those of the Cenomanian and Turonian, because the latter is so rare. However, tests for differences in size disparity between terrestrial squamates in the Aptian–Albian bin and the well-sampled Campanian bin reveals no significant difference in the range (observed versus expected, $p = 0.541$; electronic supplementary material, figure S4) or spread (observed versus expected,

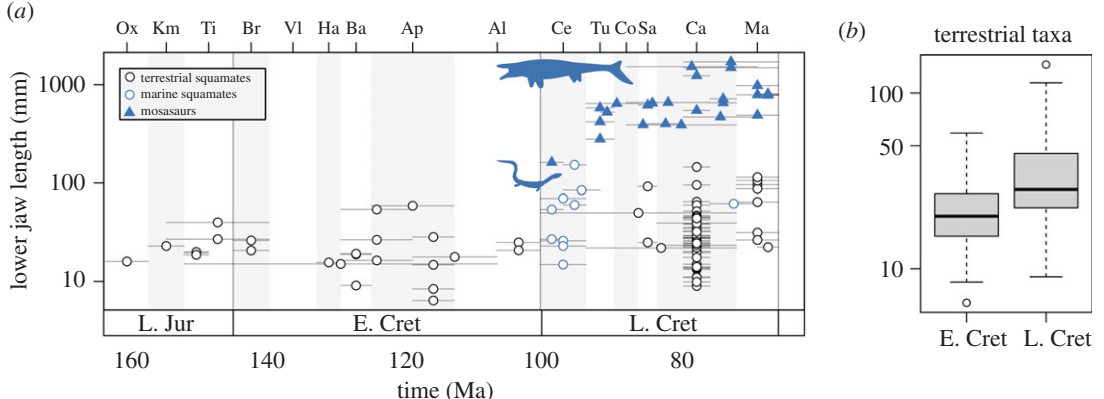

**Figure 2.** Temporal trends of early squamate size evolution, based on mandible lengths ($n = 116$). (*a*) Data are plotted at the stratigraphic midpoint for each taxon with temporal ranges denoted by grey horizontal bars. Terrestrial (black circle), marine (blue circle) and mosasauroid (blue triangle) taxa are labelled separately. (*b*) Comparison of jaw sizes in Early and Late Cretaceous terrestrial squamates using boxplots. Note the $\log_{10}$ scaled *y*-axes in both plots.

$p = 0.561$; electronic supplementary material, figure S4) of sizes. Similarly, there is no significant difference in the range (observed versus expected, $p = 0.815$; electronic supplementary material, figure S5) or spread (observed versus expected, $p = 0.405$; electronic supplementary material, figure S5) of jaw sizes in all terrestrial Early Cretaceous squamates versus all terrestrial Late Cretaceous squamates. There is a significant increase in mean jaw size when comparing terrestrial taxa from the Early and Late Cretaceous (two sample *t*-test, $t = -2.696$, d.f. $= 73$, $p = 0.009$; figure 2*b*).

There are several ecologically distinct terrestrial squamates from the Late Cretaceous that show larger jaw sizes than those in the Early Cretaceous (figure 2). The varanoids *Estesia* and *Chianghsia* and the snakes *Sanajeh* and *Dinilysia* are larger predators with jaw lengths close to or over 100 mm, and their size probably enabled them to feed on larger prey. The extinct polyglyphanodontians were specialized herbivores that had larger jaws, including *Tianyusaurus* (87 mm) and *Polyglyphanodon* (96 mm), and total body lengths over 1 m (based on complete specimens).

## 3.3. Lower jaw morphospace trends

Morphological variation in Mesozoic squamate lower jaws is visualized in biplots of principal components 1 and 2 (figure 3). For the full sample, PC1 (35.4% of total variation) represents changes in the elongation of the biting area, the relative space for muscle attachment on the surangular and angular, and the overall robusticity of the jaws. PC2 (21.8% of total variance) reflects changes in the robusticity of the dentary, the height of the coronoid process, and the curvature of the jaw's ventral margin. When analysing just Late Cretaceous taxa, the major shape changes along axes are identical and the proportion of variance encapsulated by PC1 and PC2 is similar (38.1% and 22.1%, respectively). Full details of PC axis loadings are in extended data, electronic supplementary material, table S2.

In lower jaw morphospace, Late Jurassic squamates form a relatively small cluster if snakes are not sampled (figure 3*a*). Early Cretaceous taxa largely overlap Late Jurassic forms, but there is notable expansion along PC1 (figure 3*a*). Late Cretaceous squamates occupy a larger total morphospace that subsumes the morphospaces of the Late Jurassic and Early Cretaceous taxa (figure 3*a*). Incorporating a hypothetical 'average snake' morphotype notably expands the morphospace of Late Jurassic taxa, particularly along PC1, and marginally increases the size of Early Cretaceous morphospace. Nevertheless, morphospace occupation by Late Cretaceous taxa is more expansive along both major axes with many more disparate forms at the extremes of morphospace (figure 3*a*). NPMANOVA tests return no significant results for shifting morphospace occupation between epoch time bins, suggesting that the jaw morphospace centroid for squamates was relatively stable through the Mesozoic (see electronic supplementary material, table S3).

Disparity metrics measured from morphospace axes confirm that the Late Cretaceous was a time of high disparity (figure 3*c*). However, statistical tests show that high variance and morphospace volume in the Late Cretaceous is not significantly greater than that of the Late Jurassic and Early Cretaceous, particularly if the hypothetical 'average snake' morphotype is included within those preceding bins

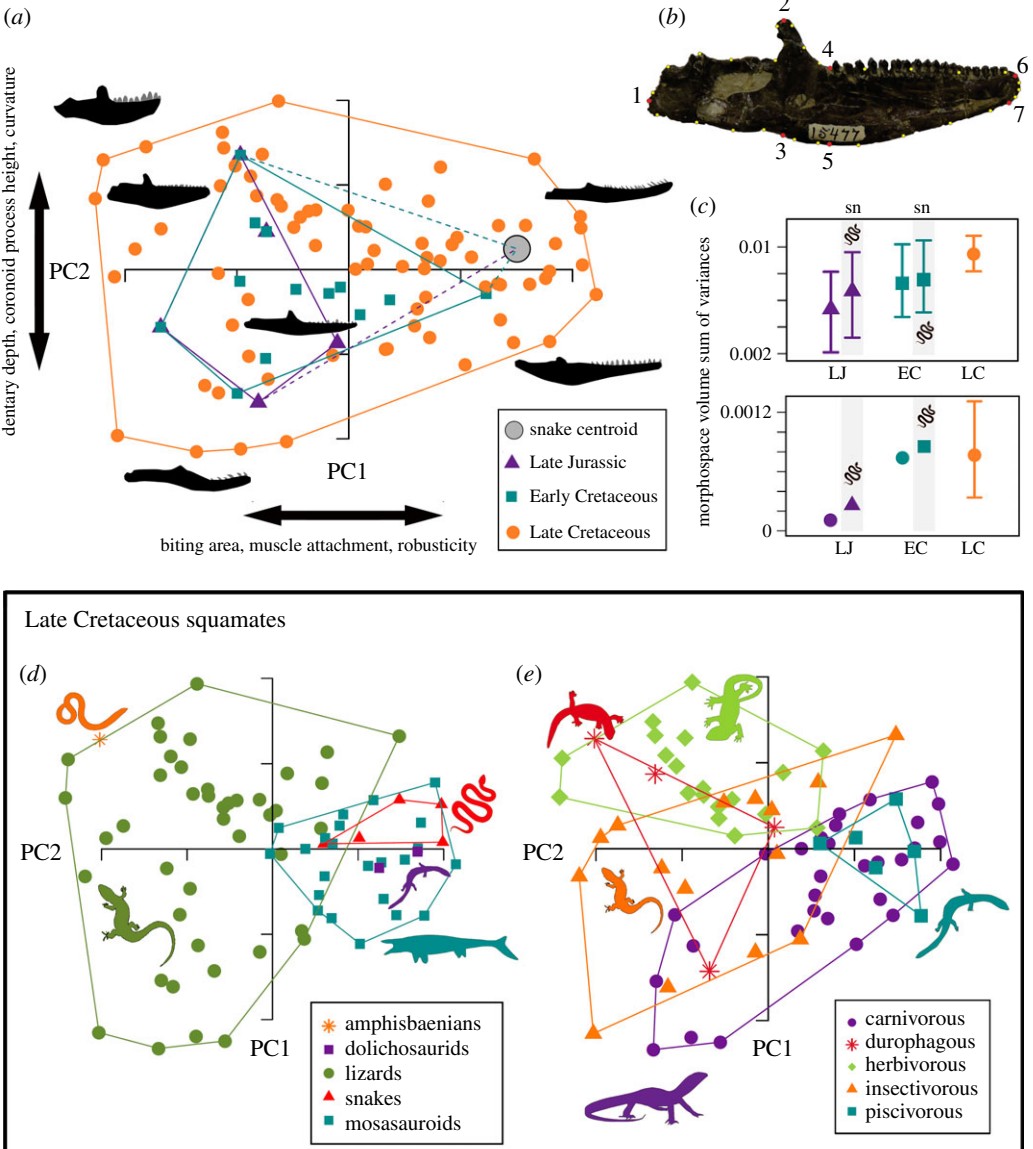

**Figure 3.** Jaw morphospaces of Mesozoic squamate genera. (*a*) Morphospace occupation from the Late Jurassic, Early Cretaceous and Late Cretaceous taxa denoted by convex hulls (*n* = 89). The convex hulls of the Late Jurassic and Early Cretaceous are expanded to incorporate the morphospace centroid of snakes (dotted lines), sampled from only the Late Cretaceous. (*b*) Landmark and semi-landmark positions illustrated on the jaw of *Polyglyphanodon sternbergi* (USNM 15477). The landmarks are (1) the most posterior point of the articular, (2) the most dorsal point of coronoid process, (3) ventral point of a vertical line from landmark 2, (4) the most posterior point of the most posterior teeth, (5) ventral point of a vertical line from landmark 4, (6) the most anterior and superior point of dentary, and (7) the most anteroventral point of dentary. Twenty-six semi-landmarks were used, and all of them are marked as yellow points. (*c*) Sum of variances and convex hull volume, with 95% error bars, are plotted for the three temporal bins, both excluding and including the snake centroid location in the Late Jurassic and Early Cretaceous bins (including highlighted in grey with 'sn'). (*d*) Morphospace of Late Cretaceous taxa divided by clades and denoted by convex hulls (*n* = 74). (*e*) Morphospace of Late Cretaceous taxa divided by dietary groups and denoted by convex hulls (*n* = 74). In (*a*), PC1 is 35.4% of total variation and PC2 is 21.8%. In (*b*) and (*c*), PC1 represents 38.1% and PC2 equals 22.1%. Morphospace occupation based on PC3 and PC4 is illustrated in electronic supplementary material, figure S6.

(figure 3*c*). Permutation tests, for significant differences in the observed disparity difference between the Early Cretaceous and Late Cretaceous compared with a null model, show no significant results, with *p*-values ranging from 0.309 to 0.928 across variance and volume metrics, and with and without the snake morphotype in the Early Cretaceous (see electronic supplementary material, figure S7). These insignificant test results reflect two things. First, the overall spread of Late Jurassic and Early Cretaceous taxa in morphospace is relatively high, given their lower sample sizes, and particularly if

the 'average snake' morphotype is included in the former bin. This means that, although Late Cretaceous squamates expand the extremities of morphospace, all four quadrants had already been partially explored previously. The second consideration is the impact of sample size on the convex hull metric. With rarefaction ($n = 13$) and bootstrapping, the mean volume value for the Late Cretaceous is similar to the Early Cretaceous (figure 3c), and if the full Late Cretaceous sample is used, the observed volume difference between the Early and Late Cretaceous is not significantly different from that expected given the sample size difference.

When the well-sampled Late Cretaceous taxa are divided into major groups, it is clear that lizards show the widest morphospace occupation, extending greatly over both PC1 and PC2 (figure 3d). Mosasauroids, dolichosaurs and snakes occupy a distinct area of morphospace restricted to positive PC1 values, with less variation on PC2. This morphotype is represented by elongated biting areas with a moderate to low jaw height and medium height coronoid processes. The only Cretaceous taxon possibly referable to a stem amphisbaenian, the 'lizard-like' *Slavoia darevskii*, represents a morphological extreme and is positioned at the extremity of PC1, close to lizards with robust jaws and very high coronoid processes such as *Adamisaurus*, *Cherminsaurus* and *Gilmoreteius* (figure 3d).

Dividing Late Cretaceous taxa into dietary guilds reveals some interesting ecological groupings (figure 3e). Carnivorous taxa have wide morphospace occupation that overlaps with insectivores and durophages, and completely subsumes the morphospace of piscivores, which form a tight cluster. Insectivorous taxa also have a wide morphospace hull and are spread across PC1 and PC2. Durophages and herbivores also have a relatively wide morphospace occupation. The wide distribution of these more specialized dietary groups shows a single morphotype that does not define them, but they do share some common features such as having more robust jaws with larger areas available for muscle attachment and with reduced tooth rows.

Comparisons of temporal, taxonomic group and dietary morphospace occupation (figure 3a,d,e) show that the expansion of jaw morphospace in the Late Cretaceous included expansions at the limits of morphospace occupied by herbivores, insectivores and carnivores. Rarer diets at this time included piscivory (uniquely among marine snakes, dolichosaurs and mosasauroids) and durophagy, which are associated with divergent jaw shapes at the extremes of PC1.

# 4. Discussion

## 4.1. Ecomorphological disparity expansion first, diversity second

Our main finding is the substantial expansion of squamate ecomorphological disparity, representing an expansion of dietary modes to near-modern levels, before the first rise in diversity of the clade in the Campanian. This is reflected by all three indices, dentition, jaw size and jaw shape, but each gives different insights into the major drivers, timings and magnitudes of ecomorphological innovation.

Squamate fossil jaws and teeth provide a reasonable proxy for dietary evolution and such remains are more frequent than other portions of the skeleton, given the rarity of complete skeletons and even complete cranial material in the fossil record. The abundance of fragmentary tooth-bearing elements provides a stage-level account of morphotype occurrences. This shows a marked expansion in the disparity of squamate dietary modes about 110–90 Ma (figure 1; electronic supplementary material, figure S1). This notably precedes the expansions of squamate diversity around 84 Ma and in the Early Palaeogene (66–55 Ma), according to fossils [3] and dated phylogenomic trees [4–7]. Taxa from the Late Jurassic to Early Cretaceous had a relatively low dental disparity, dominated by a single tooth form (simple conical teeth, 'chisel' or 'peg' like). There was a clear shift in squamate dental morphology at the end of the Albian and through the early stages of the Late Cretaceous (Cenomanian–Santonian) that ended with the high diversity and morphological disparity of Campanian–Maastrichtian taxa. Although squamate diversity continued to expand through the Cenozoic, all major dental morphologies had already been established in the Mesozoic. Therefore, we present a case of a rapid and early expansion of morphological, and potentially dietary, disparity, associated with origins of major clades, followed by a diversification of families and lower taxa 25 Myr later.

Our evidence for the expansion in dental disparity of squamates at about 110–90 Ma coincides with the time of the Cretaceous Terrestrial Revolution (KTR). As suggested before [13], it could be that the expansion in squamate feeding modes was part of the KTR, at least in North America, driven especially by the rise in insect diversity and disparity associated with the initially diversifying

angiosperms. Further, the expansion of plant resources could have been important in the diversification of the specialized herbivorous polyglyphanodontian and iguanian lizards in the Late Cretaceous [13]. Similarly, the diversification of herbivorous and insectivorous feeding modes among Mid- to Late Cretaceous spiders, birds and mammals have been linked to the diversifications of angiosperms and other plant groups, and key insect groups such as beetles, bugs, bees, ants and butterflies [12,25–29]. The increases in lizard diversity in the Late Cretaceous (84 Ma) and Early Eocene (55 Ma) post-date the onset of the KTR [4–7,27], but perhaps our finding of a feeding disparity burst 110–90 Ma connects with the initial diversification of angiosperms and diversification of terrestrial habitats, even though angiosperms then were limited to small plants and shrubs. The post-Mesozoic diversification of many squamate groups probably relates to further evolution of angiosperms and the origins of the first rainforests in the Palaeogene with all their feeding opportunities [30]. Evaluating the relative roles of these two opportunistic expansions in squamate diversity is a future area of inquiry.

The ecological diversification of squamates during the Mid-Cretaceous can be contrasted with the diversification of early mammals. The KTR has been posited as a major catalyst in early mammal evolution [26]. However, in mammals, there was a turnover of dental functional types from a heterogeneous assemblage of docodont, triconodont, symmetrodont, eupantotherian and plagiaulacoid forms in the Late Jurassic and Early Cretaceous to a more morphologically homogeneous, but probably functionally diversified, assemblage dominated by tribosphenic (primarily therian) and cimolodont (multituberculate) morphologies in the Late Cretaceous [12]. This mammal disparity turnover occurs during the KTR (approx. 125–80 Ma) and is linked to the substantial radiation of therians, but dental disparity trends are opposite of what is seen in squamate evolution.

Today, squamates show a wide range of feeding modes, but limited to the tooth types we identify here (figure 1) in Cretaceous squamates [13]. It is impossible to say truly that the breadth of diets and feeding modes today is the same as or more than in the Late Cretaceous. For example, the varanid lizards today are almost exclusively carnivores, and yet *Varanus olivaceus* feeds only on fruit, and with very little modification to its tooth shape [31]. Most modern lizards feed on insects, with iguanians specializing on ants, wasps and beetles, whereas other insectivorous lizards feed more on termites, grasshoppers, spiders and insect larvae [31,32]. This dietary division may reflect the abilities of iguanians to deal with noxious chemical defences of their prey. Gekkotans specialize by feeding largely at night, while other lizards are mainly active by day, and gekkotans and iguanians often live at higher altitudes than other lizard groups, so offering them different prey. Some such as chameleons have sticky ballistic tongues, allowing them to catch unsuspecting prey at a distance. Herbivory is much less common, seen in perhaps 2% of lizards [31,32], occurring mostly in Iguania, but also in individual species and genera in most other clades. Snakes are predators, feeding by snatching invertebrates or vertebrates and using suffocation or venom to immobilize their prey before swallowing it whole. Amphisbaenians feed on small prey and plants. Behavioural traits are difficult to assign to fossil taxa, but tooth and jaw shape, as well as phylogenetic assignment, can suggest some comparisons between Late Cretaceous and modern squamates and confirm broadly the stability of their major dietary adaptations.

The macroevolutionary patterns for Mesozoic squamate tooth disparity (figure 1) are similar to those for size changes, but marine taxa drive the latter disparity transition (figure 2). Before the beginning of the Late Cretaceous, squamates had a predominantly small body size with lower jaw lengths below 100 mm (figure 2), and they show a remarkable increase in their body size ranges into the Cenomanian and Turonian, preceding later diversity expansions. Although we excluded older taxa before the Late Jurassic (Oxfordian), two basal terrestrial squamates from the Middle Triassic and Middle Jurassic [7], had rather small body sizes based on a skull length of about 25 mm for the Triassic *Megachirella* [7,8], and about 23.5 mm for the Middle Jurassic *Marmoretta* [8]. These suggest similar sizes to those seen from the Late Jurassic and Early Cretaceous and suggest that squamates remained small through all that time span. The expansion of marine squamates, epitomized by mosasauroids, in the early stages of the Late Cretaceous represents a major evolutionary radiation [10,33], driving squamates to sizes not seen before, and not seen since the eventual extinction of mosasauroids at the end-Cretaceous. This, once again, points to a decoupling between diversity and disparity expansions.

## 4.2. Quality of the data

The squamate fossil record presents many challenges and there is a risk that we are merely documenting an apparent increase in disparity as a result of improved sampling in the Late Cretaceous, but we suggest that we have identified a real expansion in disparity in the Late Cretaceous for four reasons:

(1) Subsampling correction of squamate data through the Mesozoic and Palaeogene shows [3] that the low Jurassic and Early Cretaceous diversity levels and the high Late Cretaceous diversity levels in certain regions cannot simply be explained by bias or poor sampling. Further, we have rarefied our sample sizes and show the increases in disparity are still clear when the Late Cretaceous sample size is restricted to Early Cretaceous levels (figure 3c).

(2) The new dental morphotypes emerged through the Early Cretaceous, a time of generally poor sampling [3], well before the acknowledged improvement of sampling in the Late Cretaceous. Improved sampling through the Late Jurassic and Early Cretaceous could only add data, and so would enhance, rather than diminish, our claim for an early expansion of ecomorphospace.

(3) The increase in body size range about 100–90 Ma (figure 2) is contrary to a sampling-based explanation; larger animals are more likely to be fossilized, and so should dominate in times of supposedly poor sampling [3], and yet only small-sized specimens are found in such times. Note, however, that the largest squamates are the Late Cretaceous mosasauroids, which are expected to be well sampled [33], so this can partly explain the coincidence in the empirical and sampling-corrected peaks in diversity in the Late Cretaceous [3].

(4) Rhynchocephalians probably had a similar preservation potential to squamates, and yet they show an opposite pattern of apparent diversity, being particularly diverse in the Late Triassic, and reasonably diverse at points in the Jurassic and Early Cretaceous, before falling to very low diversity levels [3,20]. Rhynchocephalian fossils are found in similar geographic zones, in similar rock facies, and sometimes in the same localities, and their small skeletons are similar in robustness and overall size to squamates, and yet they have been found in dozens of Late Triassic and Jurassic localities, for example, and represented by tens of species. Therefore, despite patchiness of sampling through the Mesozoic, the simplest explanation for the substantially differing empirical patterns of palaeodiversity of rhynchocephalians and squamates is that patterns are broadly biological.

Trying to overcome the patchy squamate fossil record, often dominated by highly fragmentary specimens, is a challenge for future studies of morphological disparity in the group. One potentially fruitful approach would be the application of discrete characters [34–36], allowing both incomplete and complete specimens to be incorporated based on available materials, and potential subdivisions of the data to focus on ecomorphological traits. Our work explores morphological evolution based on fossil occurrences, by incorporating a phylogenetic component, and future work could explore phenotypic change along phylogenetic branches, giving deeper insights into the timing of exceptional innovation and evolutionary rate dynamics underlying these general trends.

## 5. Conclusions

Our work suggests that the initial trigger to the radiation of squamates came in the Mid-Cretaceous when they adopted the full suite of modern ecomorphological modes, perhaps linked to the explosion of angiosperms and insects comprising the KTR, and also the novel expansion of the marine mosasauroids. Further, in light of concerns about the patchy quality of the Mesozoic fossil record of squamates, improvements in the fossil record would probably either enhance the scale of the expansion of dietary ecomorphospace in the early Late Cretaceous, or shift aspects of that expansion back to older times. Our work shows that dietary ecomorphology of squamates expanded 110–90 Ma, substantially before the Late Cretaceous rise in diversity at around 84 Ma, thus confirming the decoupling of disparity and diversity, and the rise in disparity first, as seen in many other palaeontological studies [11].

Data accessibility. We provide all data for the electronic supplementary material, and at the Dryad Digital Repository: doi:10.5061/dryad.f1vhhmgvw [37].

Authors' contributions. J.A.H-F., T.L.S. and M.J.B. designed the project. J.A.H-F. collected the data. J.A.H-F. and T.L.S. performed the analysis. All authors discussed the results, wrote the manuscript and gave final approval for publication.

Competing interests. We declare we have no competing interests.

Funding. This research was funded by a PhD scholarship from CONACYT, Mexico to J.A.H-F., NERC grant NE/I027630/1 to M.J.B. and T.L.S. and ERC grant 788203 (INNOVATION) to M.J.B. and T.L.S.

Acknowledgements. We thank Max Stockdale (University of Bristol) for advice on the geometric morphometric analysis and Emily Rayfield and Susan Evans for valuable discussions and their helpful suggestions. We thank Amanda Millhouse for access to the collection of the Smithsonian Institution, and the reviewers for extremely thorough and helpful comments.

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
