## [Peer Review File · Royal Society Open Science]

Review History

RSOS-201961.R0 (Original submission)

Review form: Reviewer 1

Is the manuscript scientifically sound in its present form?

Yes

Are the interpretations and conclusions justified by the results?

Yes

Is the language acceptable?

Yes

Do you have any ethical concerns with this paper?

No

Have you any concerns about statistical analyses in this paper?

No

Recommendation?

Accept with minor revision (please list in comments)

Comments to the Author(s)

I am pleased that the authors have made efforts to improve the paper - particularly the inclusion of statistical tests of disparity differences that were absent from the original manuscript. (A side note: I don't really understand the need to say that accommodating sampling variation is 'impossible' [implied in the response letter], but that doesn't seem to have prevented the authors from inserting appropriate tests of variation in disparity through time). I also welcome changes to the text and figures that have substantially clarified the goals of the research, and the wider context.

I have a few specific comments remaining. But they constitute minor revisions. The paper can certainly be published on RSOS after those are completed. The most important concerns the treatment of angiosperm evolution - angiosperm forests were not present in the Early Cretaceous but this seems central to the mechanistic hypothesis currently proposed by the authors.

Abstract

"Until this time, squamates had uniform tooth types"

>The word 'uniform' implies all exactly the same. But this isn't the case. For example, see the occurrence of snake-like teeth in the Late Jurassic Parviraptor. Please change the language here to something that means 'low disparity'. Rather than 'uniform'.

>In the abstract, the mid-Cretaceous diversification of squamates is attributed to the "Cretaceous terrestrial revolution" and also to the diversification of the marine group (mosasauroids). These ideas seem to be at odds with each other. Please clarify. If it's diversification of marine squamates then it probably doesn't relate directly to events on land.

>The abstract is also confusing in saying "major expansion of ecomorphospace happened before... in the mid-Cretaceous" and then "the highly diverse and disparate Are Cretaceous squamates... jaw innovation in Late Cretaceous squamates involved expansions at the extreme of morphospace". These two statements seem contradictory. It sounds like you're seeing a pattern of sustained expansion of morphospace through the mid to Late Cretaceous, with includes events in both marine and terrestrial groups. You're also saying that this episode commenced well before expansions of diversity as understood from the fossil record and from molecular clocks. Could you say something like that instead of trying to pin 'expansion' to a specific point in time, which is slightly a red herring?

Introduction

>Should 'seaweed eating' be 'seagrass eating'?

"Unusually, this diversification also included a major marine group, the mosasauroids"

>Please omit the word "unusually". It isn't clear whether out is unusual for diversification in tetrapods to result in marine groups or not, or what that would mean.

"Our question is whether the fossil data document a parallel rise in dietary ecomorphological disparity through the Cretaceous, or whether diversity and disparity are decoupled [11]."

>Some context here would be helpful. Footes (e.g. 1997, review article) and others have suggested that it is quite common for disparity to increase before diversity increases. Maybe it's worth saying that, and also noting that there may be exceptions.

"Our approach is not to consider diversity through time [3]"

>Also some context here, given that the format of the journal allows sufficient space for this. Here, I think you should say what studies have already showed about squamate diversity and disparity. Or at least, say in words that diversity was already studied in (cite relevant papers). Currently a lone citation is hanging out on its own with no explanation.

"Morphological disparity can be extrapolated to provide a measure of ecomorphological variety".

>I don't think this is extrapolation but the actual definition of that word. It could be better to write "that morphological disparity can be used as a proxy for ecological variety", which is I think your intended meaning.

"... Whether diversity and disparity should have evolved in concert, perhaps a null expectation"

>If we were starting from zero empirical knowledge then this might be a reasonable default expectation. However, in this case many previous studies exist that have documented how these relationships vary in different groups and on different timescales. That body of work certainly should be indicated, and it would be useful to state any generalities or wider patterns that emerge from that. I think that prior knowledge suggests that these two often do not evolve in concert, but that this is not without exception.

Methods

"Taxa showing pronounced heterodonty were assigned two dental morphotypes in order to try to capture both possible feeding modes"

>While heterodonty does indicate differentiation of tooth functions, it seems unlikely that this should be interpreted as reflecting 'two possible feeding modes' rather than similar range of feeding modes that that in other taxa, but the modes require differentiation of tooth function. Feeding and dentitions is something we know quite a lot about but is only treated in brief here. That's OK. But it would be useful to avoid unintended meanings.

"Size disparity" and "Size evolution".

>It could be worth clarifying that mandible length is a poor proxy for body size (already stated), but a good proxy for skull size (not currently stated).

Results

"had low dental disparity, essentially comprising three morphotypes"

>The morphotypes aren't stated, unless I'm missing it. Can you state them here?

"When the well sampled Late Cretaceous taxa are divided into major clades, it is clear that lizards show the widest morphospace occupation:."

>Lizards are not a 'clade'. It might be Ok to call them a 'group'. But there are paraphyletic with respect to your other groups. This should be explain in the Methods, and you should avoid calling them a 'clade' elsewhere in the text.

Discussion

"Trends of dental disparity represent the best possible record of morphological evolution in Mesozoic squamates, given the absence of complete skeletons and even complete cranial material"

>This sentence inadvertently could be taken to imply that even if more complete remains were known, that dental disparity would still be the best record of morphological evolution. That's clearly not the case. Please changes this to say the they represent a reasonable proxy for dietary evolution, and are one of the most widely-applicable measures given the -rarity- (not 'absence') of complete skeletons and even crania...

"...we present a case of an 'early burst' of morphological, and potentially dietary, disparity"

>The 'early burst' model, as used in other works, is on in which rates of evolution are high early on in time and then decelerate. That isn't shown here. All that is shown is that an increase in disparity preceded increases in diversity. This could also result from constrained evolution under constant rates (e.g. Harmon et al 2010 'single stationary point' model - which is an Ornstein-Uhlenbeck [OU] model). I also think this was shown by Foote et al. using simulations in a paper ("Models of Morphological Diversification) in the book "Evolutionary Palaeobiology". Also see Sidlauskas et al (2007, Evolution, Carachiform fish paper).

>So it is potentially misleading to say 'early burst' here when the pattern is a pattern of early increase ind disparity with no evidence provided for high early rates.

"Similarly, the diversification of herbivorous and insectivorous feeding modes among mid- to Late Cretaceous spiders, birds and mammals have been linked to ... "

>The citations given all pertain to insects (beetles, hymenopterans) and mammals. Citations to works on spiders, bugs (hemipterans) and lepidopterans ('butterflies' here, although most are moths) and birds should be included. Or the groups should not be listed. Apologies if I'm missing a citation.

"but perhaps our finding of a feeding disparity burst 110–90 Ma connects with the initial rise of angiosperms and diversification of forest habitats. The post Mesozoic diversification of many squamate groups likely relates to further evolution of angiosperms and the origins of the first"

>The initial diversification of angiosperms was restricted to small plants/shrubs. Angiosperm tress did not appear until the Turonian (90 Ma) or thereabouts, and did not make a major contribution to firsts until much later. The authors must read the broader literature about this and refine their explanation and hypothesis. Currently it doesn't work because it plays fast and loose with the facts of angiosperm diversification. So what we gain I understanding of squamate evolution we lose in the form of a naive account of angiosperm evolution.

"To a more homogeneous assemblage dominated by tribosphenic"

>It might be worth saying 'morphologically homogeneous, but likely functionally diversified'. Also, I'd avoid saying 'homogeneous' here. For example, dental complexities among Late

Cretaceous cimolodontans range from some of the most simple morphologies indicating carnivore to complex morphologies indicating high fibre herbivory (e.g. Wilson et al 2012).

Review form: Reviewer 2

Is the manuscript scientifically sound in its present form?

Yes

Are the interpretations and conclusions justified by the results?

Yes

Is the language acceptable?

Yes

Do you have any ethical concerns with this paper?

No

Have you any concerns about statistical analyses in this paper?

No

Recommendation?

Accept with minor revision (please list in comments)

Comments to the Author(s)

On viewing this manuscript for a second time, I think that it is much improved. The narrative through the paper is stronger and the statistical testing is an excellent addition. Due to the extensive rewording throughout, I have a few further comments on phrasing which are outlined below, but otherwise I think this paper is suitable and ready to be published in RSOS.

If within space allowances, it would be beneficial to add citations for names / taxonomic descriptions where you discuss particular genera throughout.

Abstract, L4 - Changing "ecomorphospace" here to e.g. "dietary functional morphology" would perhaps introduce the study in a more accessible way.

Abstract, L10 - "in this case" could be removed.

Introduction, L5-12 - This information introduces the study rationale, which I think should come after the discussion of the squamate fossil record (L13-29). Perhaps it could be merged into the third paragraph (starting L30)?

Introduction, L16 - Perhaps "true" is more accurate than "modern-type"?

Introduction, L24-27 - Switching this around might be clearer, e.g. "Phylogenomic analyses and lineages-through-time plots show continuing diversity rise from about 84Ma and again in the Palaeogene, however the fossil record of Palaeogene squamates is sparse, likely under-sampling this biodiversity."

Methods, a, L5-6 - Presumably these fossils could not be assessed ecomorphologically because their fossils are incomplete? Could be slightly reworded to convey this.

Methods, a, L7-10 - This sentence would work better the other way around, e.g. "We chose to work at the generic level because there is little intrageneric variation in the ecomorphological traits we consider, particularly as most Mesozoic squamate genera are monospecific."

Results, 2, L14-15 - This sentence doesn't quite make sense. I assume the intention is something like "Cenomanian and Turonian terrestrial squamates are rare, making comparison with the preceding Aptian and Albian difficult when marine taxa are excluded."

Results, 2, L25-27 – What makes these animals ecologically distinct, did their size enable them to predate on novel prey?

Results, 3, L30-35 – The wording here needs to be delicately altered away from the idea that insignificant test results are an inconvenience given your hypothesis. For example, I think the word “issue” would be better as e.g. “consideration”.

Discussion – Avoid use of the phrase “of course”, it is unnecessary regardless of whether what is being discussed is considered well-known.

Discussion, 1, L62 – More accurate to phrase “behavioural traits would be hard to identify in fossils” as e.g. “behavioural traits are difficult to assign to fossil taxa”.

Discussion, 2, L10-13 – This sentence could be written more concisely, it seems to repeat the same point.

Discussion, 2, L22 – I think this statement should be softened, e.g. “Rhynchocephalians likely had a similar preservation potential...”.

Discussion, 3, L1 – Perhaps change “expansion” to “the radiation”.

Table S2 (in the files uploaded to RSOS, the file on Dryad needs updating) does not seem to have the correct data in it.

Decision letter (RSOS-201961.R0)

Dear Dr Benton

On behalf of the Editors, we are pleased to inform you that your Manuscript RSOS-201961 "Ecomorphological diversification of squamates in the Cretaceous" has been accepted for publication in Royal Society Open Science subject to minor revision in accordance with the referees' reports. Please find the referees' comments along with any feedback from the Editors below my signature.

Please submit your revised manuscript and required files (see below) no later than 7 days from today's (ie 07-Jan-2021) date. Note: the ScholarOne system will 'lock' if submission of the revision is attempted 7 or more days after the deadline. If you do not think you will be able to meet this deadline please contact the editorial office immediately.

on behalf of Prof Kevin Padian (Subject Editor)
 openscience@royalsociety.org

Associate Editor Comments to Author:

The most critical reviewers from a round of review at PRSB are now much happier, though recommend a few final tweaks to ensure your paper is ready for acceptance - please ensure you engage with and respond to these final recommendations as best you can.

Reviewer comments to Author:

Reviewer: 1

Comments to the Author(s)

I am pleased that the authors have made efforts to improve the paper - particularly the inclusion of statistical tests of disparity differences that were absent from the original manuscript. (A side note: I don't really understand the need to say that accommodating sampling variation is 'impossible' [implied in the response letter], but that doesn't seem to have prevented the authors from inserting appropriate tests of variation in disparity through time). I also welcome changes to the text and figures that have substantially clarified the goals of the research, and the wider context.

I have a few specific comments remaining. But they constitute minor revisions. The paper can certainly be published on RSOS after those are completed. The most important concerns the treatment of angiosperm evolution - angiosperm forests were not present in the Early Cretaceous but this seems central to the mechanistic hypothesis currently proposed by the authors.

Abstract

"Until this time, squamates had uniform tooth types"

>The word 'uniform' implies all exactly the same. But this isn't the case. For example, see the occurrence of snake-like teeth in the Late Jurassic Parviraptor. Please change the language here to something that means 'low disparity'. Rather than 'uniform'.

>In the abstract, the mid-Cretaceous diversification of squamates is attributed to the "Cretaceous terrestrial revolution" and also to the diversification of the marine group (mosasauroids). These ideas seem to be at odds with each other. Please clarify. If it's diversification of marine squamates then it probably doesn't relate directly to events on land.

>The abstract is also confusing in saying "major expansion of ecomorphospace happened before... in the mid-Cretaceous" and then "the highly diverse and disparate Early Cretaceous squamates... jaw innovation in Late Cretaceous squamates involved expansions at the extreme of morphospace". These two statements seem contradictory. It sounds like you're seeing a pattern of sustained expansion of morphospace through the mid to Late Cretaceous, which includes events in both marine and terrestrial groups. You're also saying that this episode commenced well before expansions of diversity as understood from the fossil record and from molecular clocks. Could

you say something like that instead of trying to pin 'expansion' to a specific point in time, which is slightly a red herring?

Introduction

>Should 'seaweed eating' be 'seagrass eating'?

"Unusually, this diversification also included a major marine group, the mosasauroids"

>Please omit the word "unusually". It isn't clear whether out is unusual for diversification in tetrapods to result in marine groups or not, or what that would mean.

"Our question is whether the fossil data document a parallel rise in dietary ecomorphological disparity through the Cretaceous, or whether diversity and disparity are decoupled [11]."

>Some context here would be helpful. Footes (e.g. 1997, review article) and others have suggested that it is quite common for disparity to increase before diversity increases. Maybe it's worth saying that, and also noting that there may be exceptions.

"Our approach is not to consider diversity through time [3]"

>Also some context here, given that the format of the journal allows sufficient space for this. Here, I think you should say what studies have already showed about squamate diversity and disparity. Or at least, say in words that diversity was already studied in (cite relevant papers). Currently a lone citation is hanging out on its own with no explanation.

"Morphological disparity can be extrapolated to provide a measure of ecomorphological variety".

>I don't think this is extrapolation but the actual definition of that word. It could be better to write "that morphological disparity can be used as a proxy for ecological variety", which is I think your intended meaning.

"... Whether diversity and disparity should evolve in concert, perhaps a null expectation"

>If we were starting from zero empirical knowledge then this might be a reasonable default expectation. However, in this case many previous studies exist that have documented how these relationships vary in different groups and on different timescales. That body of work certainly should be indicated, and it would be useful to state any generalities or wider patterns that emerge from that. I think that prior knowledge suggests that these two often do not evolve in concert, but that this is not without exception.

Methods

"Taxa showing pronounced heterodonty were assigned two dental morphotypes in order to try to capture both possible feeding modes"

>While heterodonty does indicate differentiation of tooth functions, it seems unlikely that this should be interpreted as reflecting 'two possible feeding modes' rather than similar range of feeding modes that that in other taxa, but the modes require differentiation of tooth function. Feeding and dentitions is something we know quite a lot about but is only treated in brief here. That's OK. But it would be useful to avoid unintended meanings.

"Size disparity" and "Size evolution".

>It could be worth clarifying that mandible length is a poor proxy for body size (already stated), but a good proxy for skull size (not currently stated).

Results

"had low dental disparity, essentially comprising three morphotypes"

>The morphotypes aren't stated, unless I'm missing it. Can you state them here?

"When the well sampled Late Cretaceous taxa are divided into major clades, it is clear that lizards show the widest morphospace occupation.

>Lizards are not a 'clade'. It might be Ok to call them a 'group'. But there are paraphyletic with respect to your other groups. This should be explain in the Methods, and you should avoid calling them a 'clade' elsewhere in the text.

Discussion

"Trends of dental disparity represent the best possible record of morphological evolution in Mesozoic squamates, given the absence of complete skeletons and even complete cranial material"

>This sentence inadvertently could be taken to imply that even if more complete remains were known, that dental disparity would still be the best record of morphological evolution. That's clearly not the case. Please changes this to say the they represent a reasonable proxy for dietary evolution, and are one of the most widely-applicable measures given the -rarity- (not 'absence') of complete skeletons and even crania...

"...we present a case of an 'early burst' of morphological, and potentially dietary, disparity"

>The 'early burst' model, as used in other works, is on in which rates of evolution are high early on in time and then decelerate. That isn't shown here. All that is shown is that an increase in disparity preceded increases in diversity. This could also result from constrained evolution under constant rates (e.g. Harmon et al 2010 'single stationary point' model - which is an Ornstein-Uhlenbeck [OU] model). I also think this was shown by Foote et al. using simulations in a paper ("Models of Morphological Diversification) in the book "Evolutionary Palaeobiology". Also see Sidlauskas et al (2007, Evolution, Carachiform fish paper).

>So it is potentially misleading to say 'early burst' here when the pattern is a pattern of early increase ind disparity with no evidence provided for high early rates.

"Similarly, the diversification of herbivorous and insectivorous feeding modes among mid- to Late Cretaceous spiders, birds and mammals have been linked to ... "

>The citations given all pertain to insects (beetles, hymenopterans) and mammals. Citations to works on spiders, bugs (hemipterans) and lepidopterans ('butterflies' here, although most are moths) and birds should be included. Or the groups should not be listed. Apologies if I'm missing a citation.

"but perhaps our finding of a feeding disparity burst 110–90 Ma connects with the initial rise of angiosperms and diversification of forest habitats. The post Mesozoic diversification of many squamate groups likely relates to further evolution of angiosperms and the origins of the first"

>The initial diversification of angiosperms was restricted to small plants/shrubs. Angiosperm trees did not appear until the Turonian (90 Ma) or thereabouts, and did not make a major contribution to forests until much later. The authors must read the broader literature about this and refine their explanation and hypothesis. Currently it doesn't work because it plays fast and loose with the facts of angiosperm diversification. So what we gain in understanding of squamate evolution we lose in the form of a naive account of angiosperm evolution.

"To a more homogeneous assemblage dominated by tribosphenic"

>It might be worth saying 'morphologically homogeneous, but likely functionally diversified'. Also, I'd avoid saying 'homogeneous' here. For example, dental complexities among Late Cretaceous cimolodontans range from some of the most simple morphologies indicating carnivore to complex morphologies indicating high fibre herbivory (e.g. Wilson et al 2012).

Reviewer: 2

Comments to the Author(s)

On viewing this manuscript for a second time, I think that it is much improved. The narrative through the paper is stronger and the statistical testing is an excellent addition. Due to the extensive rewording throughout, I have a few further comments on phrasing which are outlined below, but otherwise I think this paper is suitable and ready to be published in RSOS.

If within space allowances, it would be beneficial to add citations for names / taxonomic descriptions where you discuss particular genera throughout.

Abstract, L4 - Changing "ecomorphospace" here to e.g. "dietary functional morphology" would perhaps introduce the study in a more accessible way.

Abstract, L10 - "in this case" could be removed.

Introduction, L5-12 - This information introduces the study rationale, which I think should come after the discussion of the squamate fossil record (L13-29). Perhaps it could be merged into the third paragraph (starting L30)?

Introduction, L16 - Perhaps "true" is more accurate than "modern-type"?

Introduction, L24-27 - Switching this around might be clearer, e.g. "Phylogenomic analyses and lineages-through-time plots show continuing diversity rise from about 84Ma and again in the Palaeogene, however the fossil record of Palaeogene squamates is sparse, likely under-sampling this biodiversity."

Methods, a, L5-6 - Presumably these fossils could not be assessed ecomorphologically because their fossils are incomplete? Could be slightly reworded to convey this.

Methods, a, L7-10 - This sentence would work better the other way around, e.g. "We chose to work at the generic level because there is little intrageneric variation in the ecomorphological traits we consider, particularly as most Mesozoic squamate genera are monospecific."

Results, 2, L14-15 - This sentence doesn't quite make sense. I assume the intention is something like "Cenomanian and Turonian terrestrial squamates are rare, making comparison with the preceding Aptian and Albian difficult when marine taxa are excluded."

Results, 2, L25-27 - What makes these animals ecologically distinct, did their size enable them to predate on novel prey?

Results, 3, L30-35 - The wording here needs to be delicately altered away from the idea that insignificant test results are an inconvenience given your hypothesis. For example, I think the word "issue" would be better as e.g. "consideration".

Discussion – Avoid use of the phrase “of course”, it is unnecessary regardless of whether what is being discussed is considered well-known.

Discussion, 1, L62 – More accurate to phrase “behavioural traits would be hard to identify in fossils” as e.g. “behavioural traits are difficult to assign to fossil taxa”.

Discussion, 2, L10-13 – This sentence could be written more concisely, it seems to repeat the same point.

Discussion, 2, L22 – I think this statement should be softened, e.g. “Rhynchocephalians likely had a similar preservation potential...”.

Discussion, 3, L1 – Perhaps change “expansion” to “the radiation”.

Table S2 (in the files uploaded to RSOS, the file on Dryad needs updating) does not seem to have the correct data in it.

===PREPARING YOUR MANUSCRIPT===

===PREPARING YOUR REVISION IN SCHOLARONE===

Author's Response to Decision Letter for (RSOS-201961.R0)

See Appendix A.

Decision letter (RSOS-201961.R1)

Dear Dr Benton,

It is a pleasure to accept your manuscript entitled "Ecomorphological diversification of squamates in the Cretaceous" in its current form for publication in Royal Society Open Science.

Best regards,

on behalf of Professor Kevin Padian (Subject Editor)
openscience@royalsociety.org

Dear Dr Benton

On behalf of the Editors, we are pleased to inform you that your Manuscript RSOS-201961 "Ecomorphological diversification of squamates in the Cretaceous" has been accepted for publication in Royal Society Open Science subject to minor revision in accordance with the referees' reports. Please find the referees' comments along with any feedback from the Editors below my signature.

We have made all requested revisions and provided detailed point-by-point responses below.

Please submit your revised manuscript and required files (see below) no later than 7 days from today's (ie 07-Jan-2021) date. Note: the ScholarOne system will 'lock' if submission of the revision is attempted 7 or more days after the deadline. If you do not think you will be able to meet this deadline please contact the editorial office immediately.

on behalf of Prof Kevin Padian (Subject Editor)
openscience@royalsociety.org

Associate Editor Comments to Author:

The most critical reviewers from a round of review at PRSB are now much happier, though recommend a few final tweaks to ensure your paper is ready for acceptance - please ensure you engage with and respond to these final recommendations as best you can.

Yes, indeed. We do that and annotate our responses below.

Reviewer comments to Author:

Reviewer: 1

Comments to the Author(s)

I am pleased that the authors have made efforts to improve the paper - particularly the inclusion of statistical tests of disparity differences that were absent from the original manuscript. (A side note: I don't really understand the need to say that accommodating sampling variation is 'impossible' [implied in the response letter], but that doesn't seem to have prevented the authors from inserting

appropriate tests of variation in disparity through time). I also welcome changes to the text and figures that have substantially clarified the goals of the research, and the wider context.

I have a few specific comments remaining. But they constitute minor revisions. The paper can certainly be published on RSOS after those are completed. The most important concerns the treatment of angiosperm evolution - angiosperm forests were not present in the Early Cretaceous but this seems central to the mechanistic hypothesis currently proposed by the authors.

We are glad we were able to answer most requests for revision satisfactorily, and we comment further below on requested improvements.

Abstract

"Until this time, squamates had uniform tooth types"

>The word 'uniform' implies all exactly the same. But this isn't the case. For example, see the occurrence of snake-like teeth in the Late Jurassic Parviraptor. Please change the language here to something that means 'low disparity'. Rather than 'uniform'.

We prefer to correct this to 'relatively uniform' so we keep more user-friendly terminology in the Abstract.

>In the abstract, the mid-Cretaceous diversification of squamates is attributed to the "Cretaceous terrestrial revolution" and also to the diversification of the marine group (mosasauroids). These ideas seem to be at odds with each other. Please clarify. If it's diversification of marine squamates then it probably doesn't relate directly to events on land.

Yes, good point. We distinguish the two components by describing the rise of marine squamates as 'Late Cretaceous' rather than 'mid cretaceous.'

>The abstract is also confusing in saying "major expansion of ecomorphospace happened before... in the mid-Cretaceous" and then "the highly diverse and disparate Are Cretaceous squamates... jaw innovation in Late Cretaceous squamates involved expansions at the extreme of morphospace".

These two statements seem contradictory. It sounds like you're seeing a pattern of sustained expansion of morphospace through the mid to Late Cretaceous, with includes events in both marine and terrestrial groups. You're also saying that this episode commenced well before expansions of diversity as understood from the fossil record and from molecular clocks. Could you say something like that instead of trying to pin 'expansion' to a specific point in time, which is slightly a red herring?

We reshape these sentences further to avoid confusion, and replace previous sentences with "These events established modern levels of squamate feeding ecomorphology before the major steps in species diversification, confirming decoupling of diversity and disparity."

Introduction

>Should 'seaweed eating' be 'seagrass eating'? Oops, yes, changed.

"Unusually, this diversification also included a major marine group, the mosasauroids"

>Please omit the word "unusually". It isn't clear whether out is unusual for diversification in tetrapods to result in marine groups or not, or what that would mean.

Agreed, and this word omitted.

"Our question is whether the fossil data document a parallel rise in dietary ecomorphological disparity through the Cretaceous, or whether diversity and disparity are decoupled [11]."

>Some context here would be helpful. Foote (e.g. 1997, review article) and others have suggested that it is quite common for disparity to increase before diversity increases. Maybe it's worth saying that, and also noting that there may be exceptions.

Yes, good point. We add the phrase, “as has commonly been observed in fossil examples, when disparity commonly increases before diversity”.

"Our approach is not to consider diversity through time [3]"

>Also some context here, given that the format of the journal allows sufficient space for this. Here, I think you should say what studies have already showed about squamate diversity and disparity. Or at least, say in words that diversity was already studied in (cite relevant papers). Currently a lone citation is hanging out on its own with no explanation.

Well, we had mentioned the diversity bursts in the previous paragraph; but we now convert this to a synoptic sentence, “The diversity history of squamates has been explored before, based both on the fossil record [2,3] and on phylogenomic analyses [4–7], which show very low diversity from the Triassic to mid-Cretaceous, and then bursts of diversity in the Late Cretaceous and Paleogene.”

"Morphological disparity can be extrapolated to provide a measure of ecomorphological variety".

>I don't think this is extrapolation but the actual definition of that word. It could be better to write "that morphological disparity can be used as a proxy for ecological variety", which is I think your intended meaning.

Yes, agreed, and modified.

"... Whether diversity and disparity should evolved in concert, perhaps a null expectation"

>If we were starting from zero empirical knowledge then this might be a reasonable default expectation. However, in this case many previous studies exist that have documented how these relationships vary in different groups and on different timescales. That body of work certainly should be indicated, and it would be useful to state any generalities or wider patterns that emerge from that. I think that prior knowledge suggests that these two often do not evolve in concert, but that this is not without exception.

Quite so. Thanks to an earlier suggestion, we have already stated that fossil examples often show decoupling, so to say here that their evolution in concert is ‘perhaps a null expectation;’ seems reasonable.

Methods

"Taxa showing pronounced heterodonty were assigned two dental morphotypes in order to try to capture both possible feeding modes"

>While heterodonty does indicate differentiation of tooth functions, it seems unlikely that this should be interpreted as reflecting 'two possible feeding modes' rather than similar range of feeding modes that that in other taxa, but the modes require differentiation of tooth function. Feeding and dentitions is something we know quite a lot about but is only treated in brief here. That's OK. But it would be useful to avoid unintended meanings.

Yes, agreed. We hope this is clearer, “Taxa showing heterodonty were assigned to more than one dental morphotypes, as appropriate based on their varied tooth shapes.”

"Size disparity" and "Size evolution".

>It could be worth clarifying that mandible length is a poor proxy for body size (already stated), but a good proxy for skull size (not currently stated).

We clarify this, by rewriting, “In summary, we find that mandible length is a good proxy for skull size, but not for overall size. In the face of a fragmentary fossil record, Mesozoic squamate lower jaws are the most commonly preserved complete elements.”

Results

"had low dental disparity, essentially comprising three morphotypes"

>The morphotypes aren't stated, unless I'm missing it. Can you state them here?

These are added, namely "(simple conical; compressed, pointed and recurved; hooked and slender)"

"When the well sampled Late Cretaceous taxa are divided into major clades, it is clear that lizards show the widest morphospace occupation.

>Lizards are not a 'clade'. It might be Ok to call them a 'group'. But there are paraphyletic with respect to your other groups. This should be explain in the Methods, and you should avoid calling them a 'clade' elsewhere in the text.

Yes, changed to 'group' and checked throughout to clarify this.

Discussion

"Trends of dental disparity represent the best possible record of morphological evolution in Mesozoic squamates, given the absence of complete skeletons and even complete cranial material"

>This sentence inadvertently could be taken to imply that even if more complete remains were known, that dental disparity would still be the best record of morphological evolution. That's clearly not the case. Please change this to say that they represent a reasonable proxy for dietary evolution, and are one of the most widely-applicable measures given the -rarity- (not 'absence') of complete skeletons and even crania...

Yes, agreed. We revise this sentence to, "Squamate fossil jaws and teeth provide a reasonable proxy for dietary evolution and such remains are more frequent than other portions of the skeleton, given the rarity of complete skeletons and even complete cranial material in the fossil record."

"...we present a case of an 'early burst' of morphological, and potentially dietary, disparity"

>The 'early burst' model, as used in other works, is one in which rates of evolution are high early on in time and then decelerate. That isn't shown here. All that is shown is that an increase in disparity preceded increases in diversity. This could also result from constrained evolution under constant rates (e.g. Harmon et al 2010 'single stationary point' model - which is an Ornstein-Uhlenbeck [OU] model). I also think this was shown by Foote et al. using simulations in a paper ("Models of Morphological Diversification) in the book "Evolutionary Palaeobiology". Also see Sidlauskas et al (2007, Evolution, Carachiform fish paper).

>So it is potentially misleading to say 'early burst' here when the pattern is a pattern of early increase in disparity with no evidence provided for high early rates.

Yes, agreed. We remove this term, and replace it with the more general statement that "we present a case of a rapid and early expansion of morphological, and potentially dietary, disparity, associated with origins of major clades."

"Similarly, the diversification of herbivorous and insectivorous feeding modes among mid- to Late Cretaceous spiders, birds and mammals have been linked to ... "

>The citations given all pertain to insects (beetles, hymenopterans) and mammals. Citations to works on spiders, bugs (hemipterans) and lepidopterans ('butterflies' here, although most are moths) and birds should be included. Or the groups should not be listed. Apologies if I'm missing a citation.

We did not feel it necessary to cite one or several papers for each clade that kicked off through the Cretaceous; the list could then be some 12-15 papers to include all taxa, but several of the cited papers have good overviews of the wider impacts on terrestrial ecosystems of the KTR, and our original paper (Lloyd et al. 2008 [27]) provides a broad overview of all the clades noted.

"but perhaps our finding of a feeding disparity burst 110–90 Ma connects with the initial rise of angiosperms and diversification of forest habitats. The post Mesozoic diversification of many squamate groups likely relates to further evolution of angiosperms and the origins of the first"

>The initial diversification of angiosperms was restricted to small plants/shrubs. Angiosperm trees

did not appear until the Turonian (90 Ma) or thereabouts, and did not make a major contribution to firsts until much later. The authors must read the broader literature about this and refine their explanation and hypothesis. Currently it doesn't work because it plays fast and loose with the facts of angiosperm diversification. So, what we gain in understanding of squamate evolution we lose in the form of a naive account of angiosperm evolution.

This is what we say – note the discrimination between the ‘initial rise of angiosperms’ and the following sentence where we talk about “origins of the first tropical rainforests in the Paleogene...” We now add the further qualifier “even though angiosperms then were limited to small plants and shrubs”. Our text already makes it clear that angiosperm forests did not exist in the Mesozoic, and we emphasize this more now.

"To a more homogeneous assemblage dominated by tribosphenic"

>It might be worth saying 'morphologically homogeneous, but likely functionally diversified'. Also, I'd avoid saying 'homogeneous' here. For example, dental complexities among Late Cretaceous cimolodontans range from some of the most simple morphologies indicating carnivore to complex morphologies indicating high fibre herbivory (e.g. Wilson et al 2012).

Agreed, and revised as suggested.

Reviewer: 2

On viewing this manuscript for a second time, I think that it is much improved. The narrative through the paper is stronger and the statistical testing is an excellent addition. Due to the extensive rewording throughout, I have a few further comments on phrasing which are outlined below, but otherwise I think this paper is suitable and ready to be published in RSOS.

Many thanks. We make all your further requested changes.

If within space allowances, it would be beneficial to add citations for names / taxonomic descriptions where you discuss particular genera throughout.

We think this is a request to add author names, dates and citations to each genus name; this is not commonly done in non-systematic papers and would add an estimated 50 citations. For the moment, we do not do this, unless insisted on by the Editor.

Abstract, L4 – Changing “ecomorphospace” here to e.g. “dietary functional morphology” would perhaps introduce the study in a more accessible way. Agreed, and correction made.

Abstract, L10 – “in this case” could be removed. Agreed and corrected.

Introduction, L5-12 – This information introduces the study rationale, which I think should come after the discussion of the squamate fossil record (L13-29). Perhaps it could be merged into the third paragraph (starting L30)? Yes, great suggestion. We shift this section and thereby save some lines.

Introduction, L16 – Perhaps “true” is more accurate than “modern-type”? Yes, changed.

Introduction, L24-27 – Switching this around might be clearer, e.g. “Phylogenomic analyses and lineages-through-time plots show continuing diversity rise from about 84Ma and again in the Palaeogene, however the fossil record of Palaeogene squamates is sparse, likely under-sampling this biodiversity.” Agreed, and changed.

Methods, a, L5-6 – Presumably these fossils could not be assessed ecomorphologically because their fossils are incomplete? Could be slightly reworded to convey this. Yes, great suggestion. Reworded as “We could not include squamates before the Late Jurassic as occurrences are scarce and sporadic and the fossils are too complete to show jaw and dental characters sufficiently.”

Methods, a, L7-10 – This sentence would work better the other way around, e.g. “We chose to work at the generic level because there is little intrageneric variation in the ecomorphological traits we consider, particularly as most Mesozoic squamate genera are monospecific.” Good idea; we adopt this suggestion.

Results, 2, L14-15 – This sentence doesn't quite make sense. I assume the intention is something like

“Cenomanian and Turonian terrestrial squamates are rare, making comparison with the preceding Aptian and Albian difficult when marine taxa are excluded.” **Yes, not clear. Redrafted as: “It is difficult to compare terrestrial squamates between the Aptian and Albian on the one hand with those of the Cenomanian and Turonian, because the latter are so rare.”**

Results, 2, L25-27 – What makes these animals ecologically distinct, did their size enable them to predate on novel prey? **Yes, we add “and their size likely enabled them to feed on larger prey”.**

Results, 3, L30-35 – The wording here needs to be delicately altered away from the idea that insignificant test results are an inconvenience given your hypothesis. For example, I think the word “issue” would be better as e.g. “consideration”. **Agreed, and change made.**

Discussion – Avoid use of the phrase “of course”, it is unnecessary regardless of whether what is being discussed is considered well-known. **Agreed, and both instances deleted.**

Discussion, 1, L62 – More accurate to phrase “behavioural traits would be hard to identify in fossils” as e.g. “behavioural traits are difficult to assign to fossil taxa”. **Agreed, and modified.**

Discussion, 2, L10-13 – This sentence could be written more concisely, it seems to repeat the same point. **Too right. Now cut back to simply “The new dental morphotypes emerged through the Early Cretaceous, a time of generally poor sampling [3], well before the acknowledged improvement of sampling in the Late Cretaceous.”**

Discussion, 2, L22 – I think this statement should be softened, e.g. “Rhynchocephalians likely had a similar preservation potential...”. **Agreed and corrected.**

Discussion, 3, L1 – Perhaps change “expansion” to “the radiation”. **Yes, we can’t have people thinking about expanding squamates.**

Table S2 (in the files uploaded to RSOS, the file on Dryad needs updating) does not seem to have the correct data in it. **We checked this and found some data not updated, so we have updated everything and resubmitted the Supplementary data to Dryad.**